# “Imagine Your Career after the COVID-19 Pandemic”: An Online Group Career Counseling Training for University Students

**DOI:** 10.3390/bs13010048

**Published:** 2023-01-06

**Authors:** Andrea Zammitti, Angela Russo, Maria Cristina Ginevra, Paola Magnano

**Affiliations:** 1Department of Educational Sciences, University of Catania, 95124 Catania, Italy; 2Department of Philosophy, Sociology, Education and Applied Psychology, University of Padova, 35139 Padova, Italy; 3Faculty of Human and Social Sciences, Kore University, Cittadella Universitaria, 94100 Enna, Italy

**Keywords:** positive psychological resources, fear of COVID-19 pandemic, life satisfaction, online support program, university students

## Abstract

The COVID-19 pandemic has negatively influenced the psychological well-being of people around the world; university students have experienced feelings of fear of the COVID-19 pandemic, due to the intolerance of uncertainty, and a worsened quality of life, related to the reduction of social contacts. Scholars all around the world widely suggest the need to take care of this issue, proposing solutions to support students’ adjustment in the post-pandemic period. The literature on positive psychology and the life design approach has identified multiple psychological resources, the character strengths, that can sustain people’s life satisfaction and well-being in changing contexts, including their university experience. We proposed an online group career counseling intervention for 30 Italian university students (experimental group) to promote university students’ psychological resources and mitigate the long-term negative implications of the COVID-19 pandemic on life satisfaction. The other 30 students formed the control group. We found that the students engaged in the online group career counseling intervention evidenced, at the post-test, higher levels than the pre-test of (a) resilience, (b) subjective risk intelligence, (c) career adaptability, (d) self-efficacy, (e) optimism, (f) hope, (g) life satisfaction, and lower levels than the pre-test of (h) fears of COVID-19 pandemic. No differences at Time 1 and Time 2 were found in the control group. Implications for future research and practice will be discussed.

## 1. Introduction

The COVID-19 pandemic has affected the mental health and well-being of people around the world (for reviews, see [1,2]). An international study by Gloster et al. [3] examined the impact of the COVID-19 pandemic on the mental health of 9565 people from 78 different countries: results showed that an average of about 10% of the sample languished with low levels of mental health and about 50% had only moderate mental health. A large impact on people’s mental health has been linked to the fear of the COVID-19 pandemic (for a meta-analysis, see [4]). Fear is widely thought of as an emotion with functional and evolutionary significance, which attempts to protect an individual from danger [5]. Zammitti et al. [6] showed that fear of the COVID-19 pandemic harms spiritual well-being and flourishing and fosters the effect of negative affect on PTSD symptoms.

University students, in particular, have experienced significant fear of the COVID-19 pandemic related to intolerance of uncertainty [7] and a lack of quality of life related to the reduction of social contacts [8]. For example, a survey involving Turkish university students showed that 38% of respondents were worried about the COVID-19 pandemic [9]. Further, in the study by Kaparounaki et al. [10], Greek university students reported an increase in the levels of anxiety (42.5%), depression (74.3%), suicidal thoughts (63.3%), and a reduction in the quality of sleep (43.0%) and the quality of life (57.0%). Moreover, Kohls et al. [11] reported that having no indirect social contact one or two times a week, higher perceived stress, higher experienced loneliness, lower social support, and lower self-efficacy significantly predicted higher scores of depressive symptoms, hazardous alcohol use, and eating disorder symptoms. In Italy, Villani et al. [12] showed that increased anxiety has been associated with suffering from the inability to attend university, distance from colleagues, and inability to physically meet one’s partner. 

Scholars all around the world widely suggest the need to find immediate solutions to the mental health of university students amid the COVID-19 pandemic [13,14] and post-pandemic [15]. Supporting the mental health and well-being of all students should be a top priority in pandemic and post-pandemic times [16,17]. 

### 1.1. Positive Psychology and Life Design

The literature on positive psychology [18] and on life design [19] has identified multiple psychological resources that can sustain the life satisfaction and well-being of people in a changing context, including university students. 

Positive psychology is the field of study aimed at promoting the preventive psychological resources that enable people to experience greater levels of well-being [20]. The positive psychology approach pays attention to positive subjective experiences and positive individual traits that can improve well-being and quality of life and prevent diseases that arise when people experience a barren and meaningless life [18,21]. Positive psychology shifts the focus from pathology to positive features that make life worth living [21]. Resilience, hope, optimism, and self-efficacy are just some of the human strengths and resources that can allow individuals, communities, and societies to flourish [21,22,23].

Life design aims at providing answers to the unpredictable world of work and to the unclear career paths of the 21st century. The life design approach was born from the reflection that, nowadays, the future is not easy to anticipate due to the sudden changes in economic, social, and technological trajectories; in an uncertain society, the management of changes and transitions has become increasingly important [20]. In the construction of one’s career, the individual is not a passive spectator of everything that happens but actively acts on changes [24] to address future scenarios. The life design approach does not consider people’s life spans and stages separately; indeed, all the areas and experiences of a person’s life are relevant and must be taken into account in career construction [19,25,26]. Adopting this framework translates into the importance to pay attention to the development of personal resources [27] and the use of qualitative practices in career counseling activities [28]. 

In this way, life design and positive psychology can be qualified as complementary and integrated approaches that accompany people to the co-construction of the resources that will allow them to manage their careers.

Research in career domains has revealed a rapid growth in the number of published articles about career adaptability in recent years (for a review, see [29]). Career adaptability is “an individual’s resource for coping with current and anticipated tasks, transitions and traumas in occupational roles that, to some degree large or small, alter their social integration” [30] (p. 662). Career adaptability is positively related to outcomes such as life satisfaction [31], quality of life [32], and both general and professional well-being [33]. A study conducted on Italian and Spanish samples [34] suggested that university students who are confident in their career adaptability experience more life satisfaction because they are confident about their prospects.

Self-efficacy is the individual’s conviction about one’s abilities to mobilize cognitive resources and motivating courses of action needed to successfully execute a specific task within a given context [35]. Studies show that interventions early in students’ university careers may be effective [36] because self-efficacy is likely to be more fluid during the early stages of skills development [37,38]. Capone et al. [39] highlighted that during the pandemic outbreak in Italy, academic efficacy beliefs, satisfaction, and university sense of belonging could be considered relevant protective factors for students’ mental health, suggesting that supportive university environments foster student flourishing. 

Hope is the positive motivational state based on an interactively derived sense of success fueled by the perception of successful agency related to goals and by the perceived availability of successful pathways related to goals [40]. Hope correlates positively with several variables related to task accomplishment and well-being both in academic and work environments (for a review, see [41]) and positively influences emotional well-being among university students [42]. Moreover, a recent study showed that hope is an important driver of resilience and life satisfaction among undergraduate students [43].

Resilience is “the developable capacity to rebound or bounce back from adversity, conflict, and failure or even positive events, progress, and increased responsibility” [44] (p. 702). Resilience is not innate but an upgradeable ability, and it is associated with positive indicators of university students’ well-being: for example, it positively influences life satisfaction [45] and, in addition to reducing perceived stress, it promotes life satisfaction among students [46]. Moreover, Vinkers et al. [47] discussed the urgent need for a focus on resilience during the coronavirus pandemic, as it is a pivotal resource to cope with stress and vital to stay in balance.

Optimism is significantly related to resilience in university students [48]. Optimism is a general disposition to expect good outcomes [49] and refers to the propensity to learn about the experience and to build positive future scenarios [50]. The optimism to which we refer does not coincide with the tendency to complacency linked to self-deceptive beliefs that life is pleasant in all its aspects; rather, following Schneider et al. [51], we refer to that healthy and realistic optimism, typical of those seeking to guide the direction of one’s lives and subjective experiences within the boundaries established by active engagement in life. The very recent review of Piper et al. [52] showed that including individuals’ optimism and pessimism about the future substantially increases the explanatory power of standard life satisfaction models. Consistently, Pellerin and Raufaste [42] show that optimism influences life satisfaction [53] and the inner well-being of university students.

A relatively recent construct is subjective risk intelligence, defined by [54] Craparo et al. as the capacity to effectively assess the pros and cons of a decision in situations in which not all outcomes are expected. Risk intelligence can be seen as a positive attitude towards uncertainty and, as the life design paradigm suggests, with today’s instability and complexity of career paths, this ability to manage the risks associated with those paths can have a relevant influence on how people imagine and construct their professional future [55]. Among college students, risk intelligence affects how satisfied they feel about the chosen degree program (choice satisfaction) and how satisfied they feel about the degree course in terms of the utility of the competencies developed the future work, the improvement in realizing job aspirations and the positive effect on the future career (utility satisfaction; [55]). Subjective risk intelligence is influenced by values [56]. About the relationship between risk intelligence and other positive outcomes for individuals, Craparo et al. [54] suggested that future studies could explore the relationship between risk intelligence and positive outcomes for individuals, such as life satisfaction.

In summary, combining positive psychology and life design approaches can be useful in helping university students build their careers through the development of different personal resources, such as career adaptability, self-efficacy, hope, resilience, optimism, risk intelligence, life satisfaction.

### 1.2. Online Career Counseling 

Given the effects of the COVID-19 pandemic on the mental health and well-being of college students (e.g., [7,8,12]), there is a need to also carry out online career counseling activities with their benefits.

Online career counseling interventions are not a completely new practice [57], even though they have been rarely used, before the pandemic, and their management was similar to the traditional onsite experiences [58,59,60]. The use of technology in career counseling has grown in recent years [61], due to its benefits: the ability to expand services to all individuals more quickly, the constant access and immediate response by counselors, the services as a generator of career information gathering and the use of audio-visuals to enhance interventions [62]. Furthermore, interventions can be conducted at any time of the day, they can be easily recorded and evaluated in a standardized way [63]. Another advantage could be to assure lower prices and higher accessibility [64]. 

Among the disadvantages related to the use of the internet in career counseling intervention, it is possible to trace the difficulty in collecting the non-verbal aspects of the interview; moreover, this type of intervention is generally aimed at people who have specific characteristics, it is not projected on the basis of individual needs [62]. 

Online career counseling experiences based on the life design approach [65,66,67] and on positive psychology [68,69] have proved their efficacy. A recent study suggested that online programs, such as low-threshold support programs and self-management interventions, can be promising, easy to implement, and evidence-based for promoting mental health during and post-pandemic [11]. Moreover, online group career counseling interventions have already shown their effectiveness with Italian adolescents [66,69], Italian unemployed young adults [65], and also with Iranian [70] and Italian [67] university students. Consistently, in two life-design-based online career counseling interventions [64,70] in which online and face-to-face career counseling were compared, it was seen that online interventions can be more effective than traditional ones to promote career adaptability and life satisfaction in the Italian context [64] and that both online and face-to-face counseling can have significant effects on dimensions of career development in university students [70].

### 1.3. Purpose of the Research

Within the life design paradigm, the question that career counselors aim to ask is “how can we help our clients in coping with the changing world of work?” [19]. To answer this question, we have structured an online intervention, suitable for the working conditions that have spread during and immediately after the pandemic. Specifically, this research was guided by the questions: do university students’ personal resources increase after participation in an online career counseling training?

Our aim was to describe the development and application of an online career guidance intervention called “Imagine your career after the COVID-19 pandemic”. The intervention included online sessions and was aimed at a group of university students who found themselves planning their careers during the COVID-19 pandemic in Italy (starting from March 2020). In particular, we hypothesized that students engaged in the online group career counseling intervention would show, at the post-test, higher levels than the pre-test of (a) resilience, (b) subjective risk intelligence, (c) job adaptability, (d) self-efficacy, (e) optimism, (f) hopefulness, (g) life satisfaction and low pre-test levels of (h) fears of the COVID-19 pandemic. Furthermore, we assumed that we would not detect significant changes in these personal resources between the pre-test and post-test of the control group.

## 2. Methods

### 2.1. Design, Participants, and Procedure

The study involved 60 university students. Thirty students (3 males and 27 females) aged between 20 to 26 years (M = 22.80; SD = 1.79), formed the experimental (training) group and participated in an online career intervention based on the life design approach and positive psychology. Another 30 students (8 males and 22 females) aged between 19 to 27 years (M = 22.53; SD = 2.11) formed the control group.

The sample size for this study was calculated using G*power software [71,72]. Assuming an acceptable alpha error of 0.05 and aiming for 80% power, the minimum sample size was found to be 24, considering an effect size of 0.30. Having a control group and an experimental group made up of 30 and 30 students, respectively, we believe that the sample size can be considered valid.

The students of the experimental (training) group were invited to take part in a pilot training for the enhancement of personal resources; they were recruited with a convenience sampling, as they were attending their internship with one of the authors. The number of participants derived from the availability of the students to participate. In exchange for their participation, they would have received some credits for the completion of the internship planned by the degree course. All the students attended a psychology degree course. Participation in the training was entirely voluntary and each student could withdraw whenever he or she wanted. However, all the students participated in the training, and none dropped out before finishing the training. All the students signed informed consent for participation in the research.

The students in the control group were invited to participate in a longitudinal research study involving the administration of a two-stage research protocol. These students were recruited in the same university, through a pairing procedure with the training group regarding the variables measured at T1; they participated voluntarily and signed a consent to participate in the research. However, the control group received nothing in return for participation.

This study was conducted in compliance with the guidelines of the code of ethics of the Italian Association of Psychology (AIP) and Italian Society for Career Guidance (SIO); the research survey was approved by the internal review board for research in psychology of one of the universities involved (number of approval: UKE-IRBPSY-04.21.04).

### 2.2. Measures

To evaluate the effectiveness of the training, a two-stage protocol was administered, before (Time 1) and after (Time 2) the training. The same time has elapsed between T1 and T2 in the control group. 

*Biographical Section*. In this section, participants were asked to indicate age and gender.

*Multidimensional Assessment of COVID-19 pandemic-Related Fears* (MAC-RF; [73]). This was the only validated scale in Italian at the time of the training start to evaluate COVID-19 pandemic fears. This scale is composed of 8 items divided into four domains of fear (body, cognitive, interpersonal, and behavioral), mainly using a clinical point of view; we used only the two items of the interpersonal domain. The two items regarded the fear of others (“I fear that people who are around me can infect me”) and the fear for others (“I am frightened about my family members or close friends being in contact with other people and becoming infected with the coronavirus”). The answers must be given on a five-point Likert scale from 0 (*very unlike me*) to 4 (*very like me*). The other domains were clinics and did not fit the purpose of the present study. Cronbach’s alpha of the interpersonal domain was 0.69 at pre-training and 0.79 at post-training.

*Design my Future* (DMF; [74]). The instrument is composed of 19 items that evaluate future orientation and resilience. We used only 11 items to assess resilience (e.g., “I do not discourage a lot easily after a failure”). The items consist of some statements and the answers are given using a scale from 1 (*it does not describe me at all*) to 5 (*it describes me very well*). Cronbach’s alpha was 0.77 at pre-training and 0.89 at post-training.

*Subjective Risk Intelligence Scale* (SRIS; [54]). This scale has a total of 21 items that describe behaviors (e.g., “the uncertainty about possible developments of a situation paralyzes me”). Participants are required to answer on a 5-point Likert-type scale from 1 (*totally disagree*) to 5 (*totally agree*). A total subjective risk intelligence score was used. Cronbach’s alpha was 0.83 at pre-training and 0.90 at post-training.

*Career Adaptability Inventory* (CAAI; [32]). The Italian version of the Career Adapt-Abilities Scale consists of 24 items on a 5-point Likert scale from 1 (*not strong*) to 5 (*strongest*). The scale measures a score of career adaptability, and the participant should indicate what they review in the proposed statements (e.g., “Preparing for the future”). The Cronbach alpha indices of the study sample were 0.93 at pre-training and 0.95 at post-training.

*General Self-efficacy Scale* [75,76]. This is a unidimensional scale, composed of 10 items that evaluated self-efficacy on a 5-point Likert scale from 1 (*totally disagree*) to 5 (*totally agree*) (e.g., “I can solve most problems if I try hard”). Cronbach’s alpha was 0.81 at pre-training and 0.92 at post-training.

*Satisfaction With Life Scale* (SWLS; [77]). This scale measures general life satisfaction using a 7-point Likert scale from 1 (*strongly disagree*) to 7 (*strongly agree*). The 5 items that make up the scale represent statements with which the participants must indicate their degree of agreement or disagreement (e.g., “In most ways, my life is close to my ideal”). Cronbach’s alpha of the study sample was 0.91 at post-training and 0.92 at post-training.

*Visions About Future* (VAF; [78]). This instrument consists of 19 items and assesses hope, optimism, and pessimism. We used six items to assess optimism (e.g., “Usually, I am full of enthusiasm and optimism”) and seven items to assess hope (e.g., “In the future I will do what I’m not able to do today”). Responses are given using a 5-point Likert-type scale ranging from 1 (*it does not describe me at all*) to 5 (*it describes me very well*). For this study, Cronbach’s alpha was, respectively, for optimism and hope, 0.90 and 0.91 in the pre-training and 0.91 and 0.90 in the post-training.

### 2.3. Training 

The training group was involved in an online group career counseling intervention that lasted 45 days and included synchronous meetings with a career counselor and thirteen online activities divided into two sections. The description of the meetings and sections is as follows. 

During the first meeting, conducted in synchronous mode, the purpose of the training was presented, and the group was formed: each participant introduced himself/herself to the other members. In this phase, a virtual class was created, and the participants experienced the functions of the virtual class through an example activity. 

The first section included eight activities and was mainly based on time (past, present, and future). The starting point was the theme of *change* and how things can unexpectedly change (exercise 1); the reference to the pandemic situation was evident. The title of this exercise was “The present situation” and required participants to make reflections between the pre-pandemic condition and the present one. The participants then reflected on the changes they experienced in the past and brought them to today and reflected on *time management*. The *time cake* [79] was used at this point of the program (exercises 2 and 3). These two activities involved the participant creating a cake of the activities performed during the last week and comparing it with an “ideal” activity cake. In addition, the students were asked to reflect on what actions to take to make the “current” cake as close as possible to the “ideal” one. This activity, combined with the videos presented and the stimulus questions, could influence the participants’ self-efficacy and hope for a profitable future. In subsequent activities, students reflected on other *passages* between the past and present (exercise 4) and on how *fears* have been dealt with in the past (exercise 5). One of the sources of self-efficacy is past experience and reflecting on past successful experiences could positively influence future experiences. Subsequently, the students carried out some reflexive exercises on the future: first, they analyzed what *personal resources* could be useful to face situations in the future (exercise 6), then they reflected on how much and how their *future* may depend in part on the choices that are made in the present (exercise 7). These two exercises induced participants to be curious about the external environment and to feel responsible for the choices they make, also considering the fact that individuals can, in part, control their own future. These, together with self-efficacy, are dimensions of career adaptability. Exercise 8, on the other hand, proposed *reflections* on the importance of doing a job that is satisfying and how much this can affect people’s happiness. Again, a positive vision of the future was proposed to the students, trying to stimulate hope.

After the first 8 exercises, the students participated in a second meeting with the career counselor. During this second meeting, everyone provided feedback on the training and the group discussed the issues that emerged, under the supervision and guidance of the career counselor. During this meeting, the relationship between the activities and resources of the course was emphasized, reaffirming the overall objective, which was to foster the building or strengthening of personal resources to face the future.

The second section focused on personal resources: participants reflected on the positive things in their lives and the importance of learning from their *experiences* (exercises 9 and 10). In these two activities, there was a clear reference to the dimension of optimism, which refers to the propensity to learn about the experience [50]. Guided by stimulus questions and the videos, the students reflected on this important dimension. In the following two activities, the participants were required to identify their own *resilience model* (exercise 11), and then they described complex situations faced in their life during which they were shown to *have been resilient* (exercise 12). The *resilience model* is an activity that involves thinking about people with high resilience and confronting them with stimulus questions. These two activities were related to the resilience and risk intelligence dimensions. In addition, this type of exercise was demonstrated to be effective in enhancing self-efficacy [80]. The final exercise involved drawing up your own *professional project* through guiding questions (exercise 13). The students, during this activity, put in writing all their goals for the future. The activity was supported by some guidelines that helped the students write down their plans for the future.

The second section was followed by a synchronous meeting (third meeting). Again, the participants expressed their feedback on the general path and discussed, on a voluntary basis, their professional project that emerged with the final activity. Table 1 summarizes sections, the title of exercises, and activities.

### 2.4. Data Analysis

Data analysis was conducted using SPSS 25.0 statistical analysis software. 

Before conducting the analysis, we tested whether there were any differences between the control group and the experimental group.

To assess changes in the dimensions considered, over time and as a function of treatment condition, eight mixed-effects ANOVAs were used, with one between-groups factor (treatment condition) and one within-groups factor (time), as suggested by Fitzmaurice et al. [81]. The partial eta squared (*η*^2^_p_) was used to assess the effect size; this index assesses the percentage of variance explained by each dimension and is an effect measure used in educational research [82]. The reading of this index involves considering the effect small, moderate, or large according to the following threshold values: 0.01, 0.06, and 0.14 [83].

## 3. Results

### 3.1. Pre-Test Data

Before proceeding with the analysis, we checked whether there were any differences between the two groups regarding gender, age and the dimensions considered. This analysis confirms that there are no differences between the two groups before the beginning of the training (pre-training), and, therefore, that the two groups are equivalent in gender, age and starting resources. No differences were found for gender (*χ*^2^_(1)_ = 2.78, *p* = 0.10), age (*t*_(58)_ = 0.53, *p* = 0.60), resilience (*t*_(58)_ = −1.99, *p* = 0.05), fear of COVID-19 pandemic (*t*_(58)_ = 1.60, *p* = 0.12), risk intelligence (*t*_(58)_ = −1.50, *p* = 0.14), career adaptability (*t*_(58)_ = −1.57, *p* = 0.12), self-efficacy (*t*_(58)_ = −1.94, *p* = 0.06), satisfaction (*t*_(58)_ = −0.34, *p* = 0.73), optimism (*t*_(58)_ = 0.84, *p* = 0.54), and hope (*t*_(58)_ = −1.93, *p* = 0.06) (see Table 2).

### 3.2. Training Effects 

Mixed-effects ANOVAs carried out showed an interaction effect on resilience: Wilks’ Λ = 0.924, *F*(1) = 4.756, *p* = 0.03, *η*^2^_p_ = 0.076; fear of COVID-19 pandemic: Wilks’ Λ = 0.883, *F*(1) = 7.705, *p* = 0.007, *η*^2^_p_ = 0.117; risk intelligence: Wilks’ Λ = 0.795, *F*(1) = 14.940, *p* < 0.001, *η*^2^_p_ = 0.205; career adaptability: Wilks’ Λ = 0.808, *F*(1) = 13.768, *p* < 0.001, *η*^2^_p_ = 0.192; self-efficacy: Wilks’ Λ = 0.794, *F*(1) = 15.090, *p* < 0.001, *η*^2^_p_ = 0.206; optimism: Wilks’ Λ = 0.855, *F*(1) = 9.859, *p* = 0.003, *η*^2^_p_ = 0.145; hope: Wilks’ Λ = 0.871, *F*(1) = 8.596, *p* = 0.005, *η*^2^_p_ = 0.129. Tests of within-subjects indicated that for these dimensions, the overall effect of time was significant within the experimental group, but not within the control group. Specifically, in the post-test, the experimental group showed higher levels of resilience, risk intelligence, career adaptability, self-efficacy, optimism, and hope than the control group, showing that the training was effective in increasing the aforementioned personal resources. In post-test, the experimental group showed low levels of fear of COVID-19 than the control group, showing that the training was effective in decreasing the levels of fear of COVID-19. There were no differences in the extent of satisfaction between the two groups.

## 4. Discussion

The present study aimed at presenting the content and results of a student-centred online group career counseling intervention based both on life design and positive psychology approaches and aimed at the enhancement of personal resources for promoting career development and life satisfaction in the post-pandemic in a group of 30 university students. This intervention is an attempt to answer the call to conduct interventions supporting the mental health and well-being of university students in the COVID-19 pandemic and post-pandemic times [13,14,16,17]. 

In our study, the experimental group after the training showed significant improvements in the psychological resources of (a) resilience, (b) risk intelligence, (c) career adaptability, (d) self-efficacy, (e) optimism, and (f) hope. Furthermore, between the pre and post-test, (i) the fear of the COVID-19 pandemic significantly decreased. General life satisfaction (g) did not improve. In our study, the increase in life satisfaction was expected because of the increase in other personal resources; therefore, to see a significant increase in life satisfaction, maybe specific activities on this dimension are needed. A possible explanation for this unexpected result could be linked to the fact that the pandemic situation experienced was so complex as to impact the perception of one’s quality of life and the training requires a settling time to improve the perception of one’s life in relation to such significant changes in the pandemic.

In the control group, there was no improvement between the first and second administrations. This can be a further confirmation of the fact that the personal resources improvements are due to the training intervention and not to natural adaptation or time-related maturation processes.

The overall increase in the psychological resources may be due to the possibility provided by the program to positively project one’s future despite the uncertainties (b, risk intelligence), reflecting on how to cope with developmental tasks and transitions (c, career adaptability), emphasizing the sense of self-efficacy (d), and fostering the perception of being able to be successful while moving towards goals (f, hope) and the disposition to expect good outcomes (e, optimism). The whole program aimed precisely at guiding university students in the construction of their professional project and in the identification of their resilience model, which would have allowed them to focus on all those situations in which they had been to rebound or bounce back from adversity (a, resilience). Consistently, other studies showed that online courses [84] and peer support online interventions [85] seem to protect mental health and promote the well-being of students amidst the COVID-19 pandemic. These very promising results may be connected to the relevance of the socialization experiences and relationships developed during the online group career counseling intervention among students [86]. 

Mainly, these results indicated promising short-term effects of our program. Similarly, to the study by Zammitti [69], these results suggest that online career counseling intervention can be effective in lowering fear of the COVID-19 pandemic and enhancing psychological resources levels, useful to foster life satisfaction in times of a global pandemic. Consistently, Rahmadiana et al. [87] concluded that culturally adapted and transdiagnostic internet-based intervention appears to be acceptable and feasible for reducing symptoms of depression and/or anxiety and increasing the quality of life in university students in Indonesia. Our results were also in agreement with Santilli et al. [65] whose career counseling intervention showed to be effective in promoting those psychological resources that can support unemployed young adults in designing an inclusive and sustainable future, and decent work and life during the first COVID-19 lockdown in Italy.

### 4.1. Limitations

Some limitations need to be taken into consideration. First, the characteristic of the relatively small sample size limits the generalization of our results. Both the samples of experimental and control groups were composed mostly of female students; therefore, gender disparity does not allow us to compare the differences in the effects of our program between males and females. Future studies could use more gender-balanced samples. All participants attended a bachelor’s or master’s degree in psychology; hence, they probably have an additional intrinsic motivation factor for participation. Second, the use of self-report instruments can lead to memory bias. Third, we did not take into account other psychosocial processes such as flourishing, career success or university drop-out risk. 

### 4.2. Implications for Future Research and Practice

The present study provided some theoretical and practical implications for the development of university students’ life satisfaction and psychological resources in the post-pandemic context. Generally speaking, it was shown that an online group career counseling intervention for university students based on life design and positive psychology could promote several dimensions linked to career development, such as life satisfaction, resilience, risk intelligence, career adaptability, self-efficacy, optimism, and hope. 

From a theoretical point of view, combining similar approaches—in this case, life design and positive psychology—in building career counseling programs could be a useful way to address the complexity of the 21st century world of work. From a practical point of view, this study supports the evidence that online career counseling interventions [65,66,67,68,69] can promote various personal resources useful for planning one’s career and improving one’s well-being. Moreover, similarly to a previous study [69], the intervention reduced fear of COVID-19, suggesting that cultivating personal resources could reduce fear of the COVID-19 pandemic and protect individuals’ life satisfaction among university students.

Future studies could analyse the long-term effects of the online group career counseling intervention in a larger sample in a randomized controlled design, and also evaluate the effects on university students with drop-out risk. Furthermore, it could be interesting to evaluate the effects of this program using interrupted time series designs—the “next best” approach for dealing with interventions when randomization is not possible [88]—in order to estimate the effect of the training, highlighting the trend of baseline pre-treatment data and comparing it with post-treatment data.

## 5. Conclusions

This study aimed at discussing and evaluating the short-term effects of an online group career counseling training aimed at promoting several personal resources among Italian university students. Numerous authors have argued that promoting people’s positive resources during a pandemic and in the post-pandemic phase could be a key to preventing psychological distress [89], adapting to changed circumstances [5], and to protect individuals’ well-being [6,42]. Moreover, several career counseling interventions have already proved their efficacy in promoting dimensions related to career development [65,66,67,68,69,70]. The most remarkable results of the present study regard the improvement of life satisfaction and several psychological resources such as resilience, risk intelligence, career adaptability, self-efficacy, optimism, and hope, in the intervention group compared with the wait-list control group. Overall, the present study suggests that online group career counseling interventions based on both the life design paradigm and the positive psychology paradigm could be effective in promoting a set of psychological resources that can help university students to build their careers in the unpredictable world of work in the 21st century.

## Figures and Tables

**Table 1 behavsci-13-00048-t001:** Description of the training.

Section	Title of Exercise	Activities	
Introduction			
	1. The present situation	Video +	Pre-pandemic and present condition
	2. Time management		Time cake (current)
	3. Time management		Time cake (ideal)
	4. Change		Past-present passages
	5. Fear		Reflection on past situations
	6. Future		Personal reflections
	7. Future		Dreams list
	8. Happiness		Personal reflections
Feedback			
	9. Be in a good mood	Video +	Reflections on good humour
	10. Love for yourself		Reflections on optimistic people
	11. Resilience		Identification of a resilience model
	12. Risk intelligence		Situations of risk
	13. Work		Definition of the project
Closure			

**Table 2 behavsci-13-00048-t002:** Means and standard deviations of experimental and control groups at pre-test and post-test.

Measure	Experimental Group				Control Group			
	Pre-Training (T1)		Post-Training (T2)		Pre-Training (T1)		Post-Training (T2)	
	M	SD	M	SD	M	SD	M	SD
Resilience	3.61	0.43	3.84	0.67	3.86	0.52	3.80	0.53
Fear of COVID-19 pandemic	3.55	1.02	3.13	1.07	3.17	0.82	3.00	0.97
Risk Intelligence	3.17	0.43	3.41	0.59	3.34	0.44	3.37	0.54
Career Adaptability	4.11	0.48	4.41	0.49	4.30	0.47	4.25	0.54
Self-efficacy	3.57	0.49	3.91	0.61	3.80	0.41	3.97	0.56
Satisfaction	4.55	0.91	4.79	0.90	4.34	1.46	4.53	1.38
Optimism	3.32	0.73	3.63	0.74	3.44	0.82	3.53	0.91
Hope	3.74	0.60	4.01	0.66	4.05	0.64	4.04	0.62

## Data Availability

The data are available from the corresponding author upon reasonable request.

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
