# Peer review of "“Imagine Your Career after the COVID-19 Pandemic”: An Online Group Career Counseling Training for University Students"

_behavsci, 2023, doi:10.3390/bs13010048_

Round 1

Reviewer 1 Report (New Reviewer)

1.      Originality:  Does the paper contain new and significant information adequate to justify publication?: Yes. he authors identified an interesting topic and provided significant information on the relation of “Imagine your career after COVID-19 pandemic”. An online group career counseling training to facilitate university students' career through the enhancement of personal resources”. However, there are some issues that need clarification. More importantly, the discussion regarding the research gaps is missing!

2.      Relationship to Literature:  Does the paper demonstrate an adequate understanding of the relevant literature in the field and cite an appropriate range of literature sources?  Is any significant work ignored?: The literature review section is fine.

3.      Methodology:  Is the paper's argument built on an appropriate base of theory, concepts, or other ideas?  Has the research or equivalent intellectual work on which the paper is based been well designed?  Are the methods employed appropriate?: The paper is well structured and follows the standards.

4. Results:  Are results presented clearly and analysed appropriately?  Do the conclusions adequately tie together the other elements of the paper?: The results are very descriptive and contain more statistical explanation than clear application. Conclusion is not satisfactory.

5. Implications for research, practice and/or society:  Does the paper identify clearly any implications for research, practice and/or society?  Does the paper bridge the gap between theory and practice? How can the research be used in practice (economic and commercial impact), in teaching, to influence public policy, in research (contributing to the body of knowledge)?  What is the impact upon society (influencing public attitudes, affecting quality of life)?  Are these implications consistent with the findings and conclusions of the paper?: The implications part is not fine. Kindly update this section. This is very important.

6. Quality of Communication:  Does the paper clearly express its case, measured against the technical language of the field and the expected knowledge of the journal's readership?  Has attention been paid to the clarity of expression and readability, such as sentence structure, jargon use, acronyms, etc.: A professional review of the language is strongly suggested because several parts of the text are unclear.

Author Response

Dear Reviewer,

thank you for your comments.

We have tried to follow your suggestions to improve the paper.

  • We have expanded the results and conclusions sections, to make the results clearer and the conclusion satisfactory.
  • We have updated the section on implications, highlighting the implications for research and practice, trying to bridge the gap between theory and practice and underlining the connection with the findings and conclusions
  • We revised the language, to make the text clearer.

Reviewer 2 Report (New Reviewer)

This research proposes some solutions to support university students’ adjustment in the post COVID-19 pandemic through an online group career counseling intervention. The ultimate goal of this is to promote university students’ psychological resources and mitigate the long-term negative implications of COVID-19 pandemic on life satisfaction. Overall, the manuscript is interesting and timely, however, the manuscript needs some revision before full consideration of publication

Here are some suggestions for improvement

First: the title of the manuscript needs revision. The current title is too long and can be shorten

Second: the purpose of the research should be clearly stated by the end of the introduction

Third: What are the research questions of this research. They need to be discussed in the introduction as well.

Fourth: I think the first paragraph in section 1.1. lines 61-66 should be revised to more clear

Fifth: justification of the sample (why 30 students is a good sample size?) and its size should also be discussed.

Sixth: does the gender have an effect on the results since the vast majority are female? Can we compare  3 males to 27 females or 8 males to 22 females?

Seventh: lines 324 to lines 329 confirms no differences were found for gender. Please explain how you get this results

 Well done with your revised version

Author Response

Dear Reviewer,

thank you for your comments which have helped us in improving the quality of the article.

1 and 2. We have changed the title as follow: “Imagine your career after COVID-19 pandemic”. An online group career counseling training for university students.

  1. We have introduced the paragraph "1.3 Purpose of the research" and better explained the objectives and hypotheses
  2. We have revised the first paragraph in section 1.1. lines 61-66.
  3. We have included the power analysis through the G*Power software. This helps make sense of the sample size (paragraph 2.1)
  4. We have improved the limitations to the study, with reference to the imbalance for gender
  5. We have better explained the analysis.

Reviewer 3 Report (New Reviewer)

Thank you very much for the opportunity to review the manuscript entitled "Imagine your career after COVID-19 pandemic”. An online group career counseling training to facilitate university students' career through the enhancement of personal resources".

The Authors propose an online group career counseling intervention based on Life Design for 30 Italian university students aiming to promote university students’ psychological resources and mitigate the long-term negative implications of the COVID-19 pandemic on life satisfaction. The findings are satisfactory: students (experimental group) showed significant improvements in resilience, risk intelligence, career adaptability, self-efficacy, optimism, and hope.

I found the paper very interesting. The interplay between positive psychology and career construction is fascinating to me.

The authors thoroughly discuss the concepts of career adaptability, self-efficacy, hope, resilience, optimism and subjective risk intelligence. I would suggest the authors stress the link between these dimensions and the overall contribution to university students' career providing a few concluding sentences in the section "Positive Psychology and Life Design".

It would be interesting if the authors included examples of career guidance interventions based on Life Design in the Italian context with university students (not only online interventions) to provide other evidence of their effectiveness in the Italian context. Different studies have found an increase in personal resources, such as career adaptability, thanks to Life Design interventions. I was wondering if an increase in risk intelligence has been studied (and found) in previous interventions. It should be stressed as a strength of this study.

I hope that my comments are useful for authors, as they further develop the manuscript. 

Author Response

Dear Reviewer,

thank you for your comments. We have improved the paper as follows: we emphasized the link between the dimensions and "Positive Psychology and Life Design"; we included vocational guidance interventions based on Life Design in the Italian context.

Round 2

Reviewer 2 Report (New Reviewer)

the manuscript has much improved considering the comments from the first round. thanks for your revision, regards

This manuscript is a resubmission of an earlier submission. The following is a list of the peer review reports and author responses from that submission.

Round 1

Reviewer 1 Report

Dear authors,

Thank you for allowing me to revise your manuscript. I have found very interesting the development of this intervention. I think you made great work, but your manuscript could be improved with some small adjustments:

Introduction: I think you should avoid some explanations about the references (page 1, lines 32 and 36-37; page 2, line 85; page 3, lines 105, 137 and 143). Also, you should include the full spelling of PTSD acronym (page 1, line 40). I suggest maybe using the full spelling as it only appears once in the manuscript. I also detected an APA reference (page 3, line 133) that should be corrected.

Methods: the methods section is adequately explained, but it would benefit from including the type of design in the first paragraph. It will also be useful to include when the intervention was performed and the lockdown conditions (if any).

Results: I think there are well resumed, but I missed a qualitative assessment by the students about the program's usefulness and satisfaction. I understand it was not performed, so I suggest to include as a limitation. I was also surprised by the increase in each Cronbach’s alpha value in the post-test measures, and I think it should be discussed.

Discussion: as I said, I think the increasing values in each Cronbach’s alpha value in the post-test deserve a few lines.

Conclusion: I think the manuscript could benefit from a deep reflection about including this kind of intervention as a systematic offer for high education students.

I hope these suggestions can help you to improve your work.